# Impact of statins as immune-modulatory agents on inflammatory markers in adults with chronic diseases: A systematic review and meta-analysis

Solima Sabeel[1,2⊙], Bongani Motaung[1,2⊙], Kim A. Nguyen[3⊙], Mumin Ozturk[1,2], Sandra L. Mukasa[4], Karen Wolmarans[4], Dirk J. Blom[5], Karen Sliwa[6], Emmanuel Nepolo[7], Gunar Günther[7,8], Robert J. Wilkinson[9,10,11], Claudia Schacht[12], Andre Pascal Kengne[3,13], Friedrich Thienemann[4,14,15‡*], Reto Guler[1,2,9‡*]

1 Institute of Infectious Diseases and Molecular Medicine (IDM), Department of Pathology, Division of Immunology, Faculty of Health Sciences, University of Cape Town, South Africa, 2 International Centre for Genetic Engineering and Biotechnology (ICGEB), Cape Town Component, Cape Town, South Africa, 3 Non-Communicable Diseases Research Unit, South African Medical Research Council, Cape Town, South Africa, 4 General Medicine & Global Health (GMGH), Department of Medicine and Cape Heart Institute, Faculty of Health Science, University of Cape Town, South Africa, 5 Division of Lipidology, Department of Medicine and Cape Heart Institute, University of Cape Town, South Africa, 6 Cape Heart Institute and Division of Cardiology, Department of Medicine, Faculty of Health Sciences, University of Cape Town, South Africa, 7 Department of Human, Biological & Translational Sciences, School of Medicine, University of Namibia, Windhoek, Namibia, 8 Department of Pulmonary Medicine and Allergology, Inselspital, Bern University Hospital, University of Bern, Switzerland, 9 Wellcome Centre for Infectious Diseases Research in Africa, Institute of Infectious Disease and Molecular Medicine (IDM), Faculty of Health Sciences, University of Cape Town, South Africa, 10 Francis Crick Institute, London, United Kingdom, 11 Department of Infectious Diseases, Imperial College London, United Kingdom, 12 Linq management GmbH, Berlin, Germany, 13 Department of Medicine, University of Cape Town, Cape Town, South Africa, 14 Department of Internal Medicine, University Hospital Zurich, University of Zurich, Switzerland, 15 Cape Universities Body Imaging Centre (CUBIC), University of Cape Town, South Africa

‡ FT and RG also contributed equally to this work.
⊙ These authors contributed equally to this work.
* reto.guler@uct.ac.za (RG); friedrich.thienemann@uct.ac.za (FT)

## Abstract

While numerous studies have extensively documented the pleiotropic effects of statins, including their capacity to reduce inflammation, there is a lack of research estimating the anti-inflammatory effectiveness of statins among individuals with chronic diseases. This meta-analysis evaluates the effect of statin therapy on inflammatory markers and the lipid profile in patients with chronic diseases by analysing evidence from randomized controlled trials (RCTs). We conducted a systematic review and searched articles published between 1st January 1999 and 31st December 2023 in databases including PubMed, Web of Science, Scopus, and Cochrane. The meta-analysis was performed using random effects models and inverse variance. Effect measures were mean differences (MD) and 95% confidence intervals (CI). Collectively, statins significantly reduced IL-6 (*MD = -0.24* ng/dL *[95% CI, -0.36 to -0.13], $I^2$ = 98.3%, p < 0.001*), TNF-α (*MD = -0.74* ng/dL *[95% CI, -1.08 to -0.40], $I^2$ = 98.8%, p < 0.001*); and CRP (*MD = -1.58* mg/L *[95% CI, -2.22 to -0.94], $I^2$ = 86.5%, p < 0.001*).

**Data availability statement:** "All relevant data are within the manuscript and its Supporting Information files".

**Funding:** This publication was produced by StatinTB, which is part of the European and Developing Countries Clinical Trials Partnership-2 programme, supported by the European Union [grant number RIA2017T-2004-StatinTB (to S.S., B.M., M.O., S.L.M., K.W., E.N., G.G., R.J.W., C.S., F.T., and R.G.)]. This research was funded in whole, or in part, by the Wellcome Trust CIDRI-Africa [203135Z/16/Z (to R.J.W.)]. The funders had no role in study design, data collection and analysis, decision to publish, or preparation of the manuscript. There was no additional external funding received for this study.

**Competing interests:** The authors have declared that no competing interests exist.

Notably, atorvastatin demonstrated the most significant reduction in IL-6 and TNF-α levels, while fluvastatin and rosuvastatin displayed the greatest impact on decreasing CRP and LDL-C levels, respectively. Stratification by a longer treatment duration of more than four months revealed that atorvastatin achieved the most significant reduction in IL-6 and TNF-α. In conclusion, statin therapy not only regulates the lipid profile but also reduces systemic inflammatory biomarkers. Prolonged administration of statins led to a more substantial reduction in IL-6 and TNF-α, with atorvastatin exhibiting the greatest effect in our analysis.

## Introduction

Statins are lipid-lowering drugs that have been prescribed to people at high risk of cardiovascular disease (CVD) since 1987 [1]. Statins lower cholesterol biosynthesis by inhibiting 3-hydroxy-3-methylglutaryl coenzyme A (HMG-CoA) reductase [2,3]. This inhibition prevents the conversion of HMG-CoA into mevalonate, a pivotal substrate essential for cholesterol biosynthesis [4]. Despite their cardiovascular risk reduction properties, statins have shown to possess cholesterol-independent pleiotropic effects, including anti-inflammatory, anti-thrombotic, anti-oxidative, neuroprotective, anti-proliferative, plaque-stabilizing, and endothelial dysfunction-improving effects [5,6]. These non-lipid pleiotropic effects are primarily mediated via statins-mediated inhibition of protein prenylation, a multistep enzymatic process in the mevalonate pathway that regulates protein-protein interaction and cell membrane protein anchoring [7]. A growing interest in drug repurposing has prompted investigations into statins as adjunctive host-directed therapies to reduce inflammation in various infectious and non-infectious inflammatory diseases [8]. This highlights the need for additional meta-analysis studies which may provide guidance and insight for future clinical trials on selecting the most potent statin based on inflammation type. While randomized control trials (RCTs), cohort studies, and non-randomized intervention studies have collectively assessed the effectiveness of different statins in mitigating inflammation across various diseases [9–12], significant gaps remain. Specifically, the comparative potency and optimal duration of different statins as anti-inflammatory agents are poorly elucidated. The purpose of this meta-analysis of randomized controlled trials is to evaluate the anti-inflammatory effects of statin therapy in patients with chronic diseases. Specifically, this study aims to identify which statins are most effective and determine the optimal duration of treatment for maximizing their anti-inflammatory benefits. The outcomes of this meta-analysis will feed into evidence-based protocols for statin therapy, addressing lipid abnormalities and mitigating inflammatory processes in disease progression. If proven effective, clinicians may consider the varying efficacy of statins in modulating inflammation in adults with chronic diseases.

## Method

### Study design

This systematic review adhered to the guidelines set by the Preferred Reporting Items for Systematic Reviews and Meta-analyses (PRISMA) [13]. The protocol for this systematic

review has been registered and published on the International Prospective Register of Systematic Reviews (PROSPERO) database [registration number: CRD42020169919] [14]. Our review differs from the published protocol protocol [14] in two key aspects: we extended the search date and included non-randomized studies (NRS) for descriptive analysis.

## Search strategy

A comprehensive search was conducted across the PubMed-Medline, Scopus, Web of Science, and Cochrane databases to identify eligible articles using the following search terms in titles and abstracts: Statin [MeSH] OR Statins [MeSH] OR Hydroxymethylglutaryl-CoA Reductase Inhibitors [MeSH] OR HMG CoA Reductase Inhibitors OR HMG-COA OR HMG-COA OR Hydroxymethylglutaryl-CoA Inhibitors OR hydroxymethylglutaryl coenzyme a reductase OR simvastatin OR pravastatin OR fluvastatin OR atorvastatin OR rosuvastatin OR pitavastatin AND Inflammation [MeSH] OR Inflammatory [MeSH] AND randomized controlled trial [pt] OR controlled clinical trial [pt] OR randomized controlled trials [mh] OR random allocation [mh] OR double-blind method [mh] OR single-blind method [mh] OR clinical trial [pt] OR clinical trials [mh] OR ("clinical trial" [tw]) OR ((singl* [tw] OR doubl* [tw] OR trebl* [tw] OR tripl* [tw]) AND (mask* [tw] OR blind* [tw])) OR (placebos [mh] OR placebo* [tw] OR random* [tw] OR research design [mh:noexp] OR (comparative study) OR (comparative studies) OR (evaluation studies) OR (evaluation study) OR follow-up studies [mh] OR prospective studies [mh] OR controlled [tw] OR controls [tw] OR control [tw] OR prospective* [tw] OR volunteer* [tw]) NOT (animals [mh] NOT human [mh]). The "mh" was used for matching terms (MeSH term); "pt" was used to specify a publication type; "*" was used to search for a phrase that increases the sensitivity of the search and "tw" represents text words.

## Selection criteria

The inclusion criteria reported in the study protocol encompassed peer-reviewed articles published in English between 1999 and 2019. However, due to the enormous number of identified studies and delays in data extraction, analysis, and reporting, the inclusion criteria were adjusted to include studies published between 1st January 1999 and 31st December 2023. Studies were sourced from the PubMed, Web of Science, Scopus, and Cochrane databases. The literature search included randomized controlled trials (RCTs) and due to a significant number of non-randomized studies (NRS), these were included as a descriptive analysis. The search was restricted to articles investigating the impact of various statins (atorvastatin, fluvastatin, pitavastatin, pravastatin, rosuvastatin, simvastatin) on the lipid profile in conjunction with at least one of the following inflammatory markers: C-reactive protein (CRP), either measured conventionally as CRP or with a highly sensitive assay (hs-CRP), tumour necrosis factor-alpha (TNF-α), interleukin-1β (IL-1β), IL-6, IL-8, the soluble cluster of differentiation 14 (sCD14) or sCD16. Furthermore, the selection was restricted to studies involving adults with chronic diseases receiving statin treatment for a minimum of three months. The former is defined as conditions that last one year or longer and require ongoing medical treatment or limit activities of daily living [15] such as CVD, diabetes, hypertension, obesity, arthritis, Alzheimer's disease, epilepsy, asthma, chronic obstructive pulmonary disease (COPD), HIV, TB, etc. Studies that included participants with malignancies, autoimmune diseases, or genetic disorders; no placebo group or using cerivastatin and lovastatin were excluded. Studies classified as low quality by the Jadad scale or reported unsuitable outcome measures were further excluded from the meta-analysis.

## Data extraction

Two reviewers, SS and BM, independently screened titles and abstracts. Then, full texts were screened to extract relevant variables. Data extraction was conducted using predesigned tables; including three domains: (I) identification of the study (year of publication, first author's name, PubMed identification number, title, journal name, and impact factor); (II) methodology (study type, co-medication with statin intervention, target population, median/mean age, gender distribution, race, target condition, comorbidities, statin type, dosing, and duration of intervention); and (III) outcomes (change or relevant data to estimate the change in lipid profile and inflammation markers: hs-CRP, CRP, TNF-α, IL-1β, IL-6, IL-8, sCD14 or sCD16.

## Methodological quality and risk of bias assessment

The methodological quality of RCTs was assessed using the Oxford quality scoring system (Jadad scale) [16]. The Jadad scoring system consists of five pivotal questions: (I) was the study described as randomized; (II) was the method to generate a sequence of randomization appropriate; (III) was the study described as blinded; (IV) was the method of blinding appropriate; and (V) was there an appropriate description of withdrawals and/or drop-outs. Each question scores one point. A study that scored three points or more out of five on the Jadad scale was categorized as high-quality RCT [16].

The risk of bias for RCTs was evaluated by employing the RoB-2 tool [17]. The ROB-2 assessed five sources of bias: selection bias (I) due to inadequate generation of a randomized sequence and (II) due to inadequate concealment of allocations before assignment; performance bias due to knowledge of the allocated interventions by participants and personnel during the study; detection bias due to knowledge of the allocated interventions by outcome assessors; attrition bias due to amount, nature or handling of incomplete outcome data and reporting bias due to selective outcome reporting [17].

The risk of bias for non-randomized studies (observational studies) was determined by the Methodological Index for Non-randomized Studies (MINORS) score based on the MINORS checklist [18]. MINORS estimated the risk arising from (I) unclearly stated aims; (II) lack of predetermined inclusion criteria for consecutive patients; (III) absence of a protocol established before the study's commencement; (IV) inappropriateness of endpoints in line with the study's aim; (V) inadequate assessment of the study's endpoint; (VI) insufficient follow-up period to meet the study's aim; (VII) loss of follow-up; and (VIII) missing prospective calculation of the study's size. The items were scored 0 when not reported; 1 when inadequately reported, and 2 when adequately reported. The ideal global score on the MINORS scale for non-comparative studies was 16. Studies with scores of 12 or higher were judged as low risk of bias [17]. The quality of cohort studies and case-control studies was evaluated using the Newcastle-Ottawa Scale (NOS). Studies were categorized as: a score of 0–3 stars was deemed low-quality, a score of 4–6 stars was deemed moderate-quality, and a score of 7 or more stars was deemed high-quality [18]. Any disagreements were discussed until resolved. The Grading of Recommendations, Assessment, Development, and Evaluations (GRADE) approach was used to rank the certainty of evidence as it related to the studies that contributed data to the meta-analyses for the prespecified outcomes. The GRADE tool includes five considerations (study limitations, inconsistencies of results, imprecision, indirectness and publication bias) to assess the certainty of the evidence from RCTs that can be classified as high, moderate, low or very low [19]. The decisions to downgrade the quality of evidence were justified using footnotes. One review author (KAN) made judgments about the evidence certainty, and a second review author (APK) checked this evaluation.

## Statistical analysis

Meta-analysis of RCTs was performed using R version 4.0.4 (2021-02-15) and "meta" package. For each study included, the mean difference (MD) elucidating the effect of statins versus placebo was calculated across all studies and within major subgroups pertaining to each relevant outcome. MD and its 95%-confidence intervals were the effect measure of choice for continuous outcome variables. The DerSimonian-Laird random effects model was used to combine estimates from different studies to generate the overall MD across studies, according to the statins used. The random effects model was chosen over the fixed effects in anticipation of substantial variations across the included studies. Data from the intention-to-treat (ITT) analyses of trials (as reported by trial authors) were used. Heterogeneity among studies was assessed by visual inspection of the forest plots and by using $I^2$ statistics. The latter describes the percentage of total variation across studies that is due to heterogeneity rather than sampling error [20]. An $I^2$ value of <50% indicated low or moderate heterogeneity, and >75% represented considerable heterogeneity. Sources of heterogeneity were explored using pre-specified subgroup analyses when an adequate number of studies were permitted. Subgroup comparisons were performed using Q-test based on the Analysis of Variance (ANOVA). For studies presenting data as median and 25th-75th percentiles or minimum-maximum, the estimation of sample mean and standard deviation was carried out using

formulas recommended in the Cochrane Handbook (Higgins 2002) before performing the meta-analysis. Sensitivity analysis allowed for the exclusion of studies with skewed data exploring their impact, if any, on the pooled effect estimates. The presence of publication bias was explored using funnel plots supplemented by formal statistical assessments using Egger's test of bias. The difference was statistically significant if $P < 0.05$.

## Results

### Literature search

The initial database search identified a total of 7413 articles distributed as follows: PubMed 1173, Web of Science 3589, Scopus 765, Cochrane 1859, and other sources 27. Following this, 899 duplicated studies were removed, leaving 6514 unique studies to undergo rigorous evaluation against the predefined eligibility criteria outlined in the methods section. Out of these, 6436 studies were excluded due to various reasons highlighted in S1 Table. A total of 78 articles (48 RCTs and 30 NRS) met the criteria for comprehensive full-text review. While NRS articles (n = 30) were excluded from the meta-analysis, 48 NRS data were extracted as descriptive analysis in the S2 Table. Among the 48 RCT articles, 16 studies were excluded due to low quality based on the Jadad score, and 5 were excluded because their outcomes were not compatible with the meta-analysis (Fig 1). In summary, the meta-analysis incorporated data from 31 RCTs featured in 27 published articles spanning the years 2002–2021 (S3 Table). Article inclusion was performed following PRISMA guidelines as shown in Fig 1.

### Characteristics of included participants

 **I. Randomized controlled trials.** The meta-analysis comprised a total of 28059 randomized participants distributed across 31 RCTs, with an average age of 57 years. A substantial portion of the included RCTs focused on participants with diabetes (n = 7, 22.6%), and metabolic syndrome (5 RCTs, 16.1%). Included RCTs involving patients with coronary or cardiomyopathy diseases had equivalent numbers (3 RCTs, 9.7%), and RCTs involving patients with hypercholesterolemia, COPD or chronic renal failure (dialysis patients) were equivalent in numbers (2 RCTs, 6.5%). The remaining RCTs (7 RCTs, 22.6%) encompassed single studies related to diverse conditions such as obstructive sleep apnea, rheumatoid arthritis, polycystic ovary syndrome, elevated CRP, hypertension, schizophrenia, and HIV (S3 Table). Among the 31 RCTs, 6.5% (2 RCTs) included international populations [21,22], 35.5% (11 RCTs) were conducted in Europe [9,11,23–31], 25.8% (8 RCTs) were conducted in Asian populations [32–37], 29% of studies (9 RCTs) took place in USA [38–44], and one RCT (3.2%) was carried out in Latin America [45]. Regarding treatment duration, the majority (60%) of studies administered statins for 3–4 months, leading to the adoption of this timeframe as a cutoff point for duration assessment. Regarding the specific statins prescribed, atorvastatin was used in 17 studies (54.8%), simvastatin in seven studies (22.6%), rosuvastatin in four studies (12.9%), fluvastatin in two studies (6.5%), and pravastatin in one study (3.2%) (S3 Table).

 **II. Non-randomized studies.** A total of 1630 participants in 48 NRS from 30 published articles were examined for lipid profile and inflammatory markers (S2 Table). Thirty-eight studies (79%) were cohort studies while ten studies (21%) were case-control studies. Participants received atorvastatin (762), simvastatin (277), rosuvastatin (106), pitavastatin (310), fluvastatin (80), and pravastatin (95). Studies were conducted on patients with cardiovascular diseases, metabolic disorders, and renal diseases. All the studies measured changes in lipid profile mainly as percentage lowering of LDL-C and at least one of the selected inflammatory markers.

### Assessment of methodological quality and risk of bias of the included studies

A total of 48 RCTs were assessed for the quality of methodology using Jadad scoring (S4 Table). Thirty-two studies (66.6%) were of high quality (scores ≥ 3), whereas 16 studies (33.3%) were deemed low quality (scores < 3) resulting in a total of 32 studies that were further screened to assess the risk of bias (S5 Table). Among these, five RCTs had a high risk of selection bias [24,26,31,36,37] and one study displayed an uncertain risk of bias [28]. In terms of performance bias, six

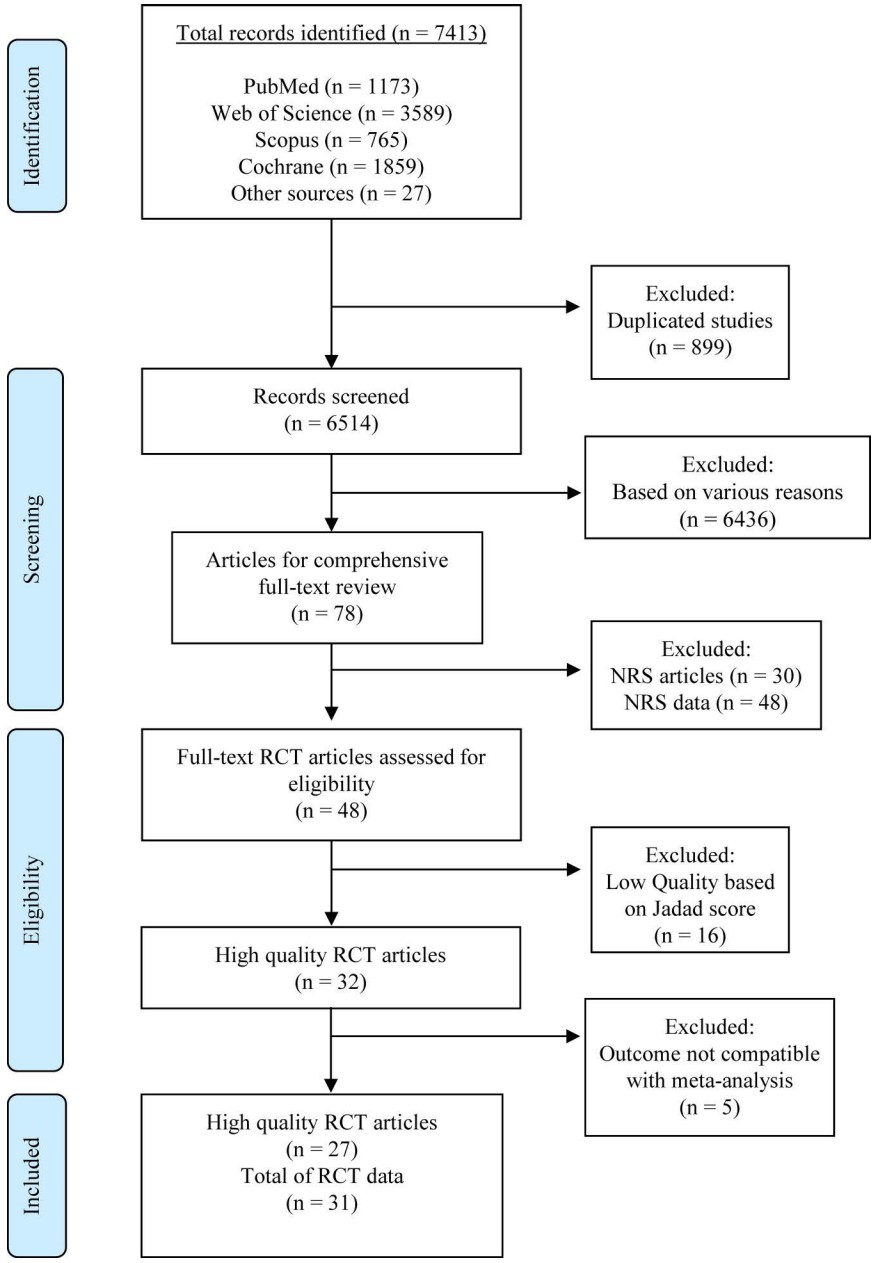

**Fig 1. PRISMA flow diagram of the systematic literature search process.** PRISMA: Preferred Reporting Items for Systematic Reviews and Meta-Analyses; RCTs: randomized controlled trials; NRS: non-randomized studies.

studies exhibited a high risk, while another six studies had an unclear performance bias [22,24,28,33,39,40]. Furthermore, four RCTs were associated with a high risk of detection bias, with another seven RCTs showing unclear detection bias. Most studies did not report additional challenges which were encountered during the trial, these were denoted as unclear in other biases (S5 Table). All identified NRS showed a low risk of bias through MINORS scoring (score = 12 or above) as highlighted in the S8 Table. Quality assessment of identified NRS through NOS showed a notable 46.7% (7 studies) which

were of high quality with a low risk of bias, while 53.3% (8 studies) showed moderate quality (S9 Table). Among the 32 high-quality RCT articles initially considered, five additional articles were excluded because their outcomes were not compatible with the meta-analysis. This led to a final selection of 27 RCT articles, all of which exhibited high quality (Fig 1).

**Impact of statins on inflammatory markers**

IL-6 was examined across seven RCTs (242 participants on statins and 241 on placebo), TNF-α was assessed in six RCTs (172 participants on statins and 171 on placebo), and CRP evaluation from 15 RCTs (4824 participants on statins and 4368 on placebo) (Table 1). Overall, statins significantly reduced IL-6 (MD = -0.24 ng/dL [95% CI, -0.36 to -0.13], $I^2$ = 98.3%, p < 0.001), TNF-α (MD = -0.74 ng/dL [95% CI, -1.08 to -0.40], $I^2$ = 98.8%, p < 0.001), and CRP (MD = -1.58 mg/L [95% CI, -2.22 to -0.94], $I^2$ = 86.5%, p < 0.001) as illustrated in Fig 2A-2C and Table 1. Overall analysis showed that atorvastatin was the most potent statin in decreasing levels of IL-6 (MD = -5.39 ng/dL [95% CI, -9.68 to -1.11], $I^2$ = 57.4%) and TNF-α (MD = -3.32 ng/dL [95% CI, -5.74 to -0.91], $I^2$ = 99.5%, p < 0.001). Fluvastatin exerted the most pronounced impact on CRP reduction (MD = -7.36 mg/L [95% CI, -8.82 to -5.91]), followed by rosuvastatin (MD = -2.07 mg/L [95% CI, -3.96 to -0.17]) as illustrated in Table 1.

The duration of statin treatment notably affected the expression of inflammatory biomarkers including IL-6 and TNF-α, as shown in Fig 3A-3B and Table 2. Statin treatment duration exceeding four months, atorvastatin decreased both IL-6 (MD = -5.39 ng/dL [95% CI, -9.68 to -1.11]) and TNF-α levels (MD = -10.20 ng/dL [95% CI, -11.20 to -9.19]), while simvastatin reduced only IL-6 levels (MD = -8.95 ng/dL [95% CI, -15.14 to -2.76]). In contrast, IL-6 and TNF-α levels were not significantly affected by 3–4 months of statin treatment. CRP levels were significantly affected by rosuvastatin (MD = -15.50 mg/L [95% CI, -31.00 to 0.00]) and fluvastatin (MD = -7.80 mg/L [95% CI, -10.17 to -5.43]) at 3–4 months of statin treatment. However, a significant reduction of CRP levels with extended statin treatment (above 4 months) was only observed with fluvastatin (MD = -7.10 mg/L [95% CI, -8.95 to -5.25]) (Fig 3C and Table 2). Stratification by age revealed significant reductions in inflammatory biomarkers, including IL-6 (MD = -5.40 ng/dL [95% CI, -12.62 to -0.91], $I^2$ = 87.5%, p < 0.001), CRP (MD = -2.02 mg/L [95% CI, -2.88 to -1.15], $I^2$ = 92.4%, p < 0.001), and hs-CRP (MD = -1.37 mg/L [95% CI, -1.82 to -0.13], $I^2$ = 75.4%, p < 0.001), among the older population (>57 years) when compared to the younger population (<57 years). Additionally, TNF-α (MD = -1.71 ng/dL [95% CI, -2.66 to -0.75], $I^2$ = 99.0%, p < 0.001) exhibited a significant reduction in the younger group (<57 years) (S10 Table).

**Certainty of evidence.** We found no high-certainty evidence to answer our research question. The certainty of evidence was generally low for IL-6 and TNF-α, and moderate for CRP and hs-CRP outcomes. Factors contributing to the reduction in certainty were inconsistency and imprecision (Table 1).

For outcomes with treatment duration longer than 4 months, the certainty of the evidence was categorised as low for IL-6 and TNF-α, and moderate for CRP and hs-CRP (Table 2). Whereas the certainty of evidence was very low for IL-6 and TNF-α, and was low for CRP and hs-CRP with the treatment duration 3–4 months. The reasons for reducing the certainty of evidence were inconsistency, imprecision, indirectness, or high risk of bias (Table 2).

A total of 48 NRS were included in the descriptive analysis, with 47 studies investigating the effects of statins on CRP and 7 studies evaluating TNF-α (S2 Table). Forty-one studies (87.2%) showed a reduction in CRP levels, two studies (4.2%) showed no change from the baseline, and four studies (8.5%) reported a slight elevation in CRP concentration following a low dose of atorvastatin treatment. One study reported a significant reduction in TNF-α following atorvastatin administration (S2 Table).

**Effect of statins on lipid profile**

The impact of statins on lipid profile is summarised in S6 Table for the overall analysis of studies incorporated in the meta-analysis. S10 Table highlights stratification by age, while Table 2 provides a stratification based on treatment duration from the included studies. Overall, thirty-one studies were included in the pooled analysis regarding the effects of statins on various lipid variables. Thirty studies reported on LDL-C reduction with a total of 28026 participants where

**Table 1. Overall analysis of randomized controlled trials.**

| Groups | Outcomes | Criteria | Studies (n) | Interven-tion (n) | Control (n) | Mean Differences (95 CI) | I² (95 CI) | P-value | P-Egger's test | Certainty of the Evidence (GRADE) |
|---|---|---|---|---|---|---|---|---|---|---|
| **Overall** | | | | | | | | | | |
| | IL-6 | Overall | 7 | 242 | 241 | -0.24 [-0.36; -0.13] | 98.3 [97.6; 98.8] | <0.001 | 0.173 | ⊕⊕⊖⊖ Low a,b |
| | | Atorvastatin | 2 | 94 | 95 | -5.39 [-9.68; -1.11] | 57.4 [0.0; 89.8] | 0.125 | NC | ⊕⊕⊖⊖ Low b,d |
| | | Simvastatin | 3 | 97 | 95 | -0.01 [-0.21; 0.20] | 75.8 [20.1; 92.6] | 0.016 | 0.581 | ⊕⊖⊖⊖ Very low a,b,d |
| | | Pravastatin | 1 | 30 | 30 | -0.25 [-1.48; 0.98] | NC | NC | NC | |
| | | Rosuvastatin | 1 | 21 | 21 | -0.00 [-0.00; 0.00] | NC | NC | NC | |
| | TNF-α | Overall | 6 | 172 | 171 | -0.74 [-1.08; -0.40] | 98.8 [98.3; 99.1] | <0.001 | 0.307 | ⊕⊕⊖⊖ Low a,b |
| | | Atorvastatin | 3 | 88 | 87 | -3.32 [-5.74; -0.91] | 99.5 [99.2; 99.6] | <0.001 | 0.102 | ⊕⊕⊖⊖ Low a,b |
| | | Simvastatin | 2 | 54 | 54 | -0.03 [-0.10; 0.04] | 83.4 [31.1; 96.0] | 0.014 | NC | ⊕⊖⊖⊖ Very low a,b,d |
| | | Pravastatin | 1 | 30 | 30 | 0.42 [-0.23; 1.07] | NC | NC | NC | |
| | CRP | Overall | 15 | 4824 | 4368 | -1.58 [-2.22; -0.94] | 86.5 [79.4; 91.2] | <0.001 | 0.274 | ⊕⊕⊕⊖ Moderate a |
| | | Atorvastatin | 6 | 1673 | 1670 | -0.80 [-1.32; -0.28] | 31.7 [0.0; 72.3] | 0.198 | 0.427 | ⊕⊕⊕⊖ Moderate b |
| | | Simvastatin | 3 | 2338 | 2310 | -0.53 [-0.66; -0.39] | 0.0 [0.0; 89.6] | 0.857 | 0.148 | ⊕⊕⊕⊖ Moderate b |
| | | Pravastatin | 1 | 30 | 30 | -1.32 [-4.78; 2.14] | NC | NC | NC | |
| | | Rosuvastatin | 3 | 759 | 338 | -2.07 [-3.96; -0.17] | 35.3 [0.0; 79.1] | 0.213 | 0.168 | ⊕⊕⊕⊖ Moderate b |
| | | Fluvastatin | 2 | 24 | 20 | -7.36 [-8.82; -5.91] | 0.0 [0.0; 0.0] | 0.648 | NC | ⊕⊖⊖⊖ Very low a,b,c |
| | hs-CRP | Overall | 16 | 9432 | 9435 | -0.81 [-1.25; -0.37] | 98.7 [98.4; 98.9] | <0.001 | 0.072 | ⊕⊕⊕⊖ Moderate a |
| | | Atorvastatin | 11 | 335 | 342 | -0.75 [-1.33; -0.18] | 94.5 [91.9; 96.3] | <0.001 | 0.028 | ⊕⊕⊖⊖ Low a,b |
| | | Simvastatin | 4 | 196 | 192 | -0.83 [-1.79; 0.14] | 66.5 [1.8; 88.5] | 0.030 | 0.936 | ⊕⊖⊖⊖ Very low a,b,c |
| | | Rosuvastatin | 1 | 8901 | 8901 | -1.30 [-1.38; -1.22] | NC | NC | NC | |

n = total numbers; NC = Not Computable; IL-6 = Interleukin 6; TNF-α = Tumor necrosis factor alpha; CRP = C-reactive protein; hs-CRP = High-sensitivity C-reactive protein.

GRADE Working Group grades of evidence:

- High certainty: we are very confident that the true effect lies close to the effect estimate.

- Moderate certainty: we are moderately confident in the effect estimate: the true effect is likely to be close to the estimate of the effect, but there is a possibility that it is substantially different.

- Low certainty: our confidence in the effect estimate is limited: the true effect may be substantially different from the estimate of the effect.

- Very low certainty: we have very little confidence in the effect estimate: the true effect is likely to be substantially different from the estimate of effect.

a: serious inconsistency (high heterogeneity, wide confidence intervals); b: serious imprecision (optimal information size (OIS) not met, small number of studies); c: serious indirectness (studied populations differed with the populations recommended); d: serious risk of bias (ROB) (if high ROB presented in half or less than half of included studies).

## A. IL-6

|  |  | Experimental | | | Control | | | Mean Difference | MD | 95%-CI | Weight |
|---|---|---|---|---|---|---|---|---|---|---|---|
| Study | Statins | Total | Mean | SD | Total | Mean | SD | | | | |
| Sola 2006 | Atorvastatin 20mg | 54 | 13.30 | 0.80 | 54 | 17.30 | 1.40 | | -4.00 | [-4.43; -3.57] | 5.9% |
| Zhang 2015 | Atorvastatin 20mg | 40 | 12.33 | 10.92 | 41 | 21.15 | 16.78 | | -8.82 | [-14.97; -2.67] | 0.0% |
| Jialal 2007 | Simvastatin 20mg | 26 | 0.03 | 0.02 | 26 | 0.04 | 0.03 | | -0.01 | [-0.03; 0.00] | 35.6% |
| Kaczmarek 2010 | Simvastatin 40mg | 28 | 0.36 | 0.28 | 28 | 0.34 | 0.30 | | 0.02 | [-0.13; 0.18] | 21.8% |
| Zhang 2015 | Simvastatin 40mg | 43 | 12.20 | 11.57 | 41 | 21.15 | 16.78 | | -8.95 | [-15.14; -2.76] | 0.0% |
| Zanetti 2020 | Rosuvastatin 10mg | 21 | 0.01 | 0.01 | 21 | 0.01 | 0.01 | | 0.00 | [-0.01; 0.01] | 35.7% |
| Vincenzi 2015 | Pravastatin 40mg | 30 | 2.93 | 1.84 | 30 | 3.18 | 2.91 | | -0.25 | [-1.48; 0.98] | 0.8% |
| Random effects model | | | | | | | | | -0.24 | [-0.36; -0.13] | 100.0% |

Heterogeneity: $I^2 = 98\%$, $\tau^2 = 0.0095$, $p < 0.01$

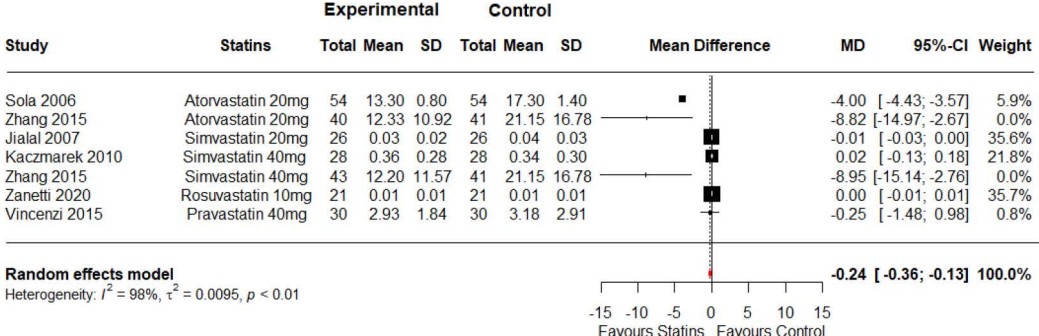

-15 -10 -5 0 5 10 15
Favours Statins  Favours Control

## B. TNF-α

|  |  | Experimental | | | Control | | | Mean Difference | MD | 95%-CI | Weight |
|---|---|---|---|---|---|---|---|---|---|---|---|
| Study | Statins | Total | Mean | SD | Total | Mean | SD | | | | |
| Sola 2006 | Atorvastatin 20mg | 54 | 24.30 | 2.30 | 54 | 34.50 | 3.00 | | -10.20 | [-11.21; -9.19] | 7.3% |
| Economides 2004 | Atorvastatin 20mg | 15 | 0.28 | 0.19 | 15 | 0.38 | 0.35 | | -0.10 | [-0.30; 0.10] | 19.6% |
| Economides 2004 | Atorvastatin 20mg | 19 | 0.42 | 0.38 | 18 | 0.46 | 0.35 | | -0.04 | [-0.28; 0.20] | 19.1% |
| Jialal 2007 | Simvastatin 20mg | 26 | 0.01 | 0.01 | 26 | 0.01 | 0.01 | | -0.00 | [-0.01; 0.00] | 21.1% |
| Kaczmarek 2010 | Simvastatin 40mg | 28 | 0.18 | 0.07 | 28 | 0.25 | 0.13 | | -0.07 | [-0.13; -0.02] | 21.0% |
| Vincenzi 2015 | Pravastatin 40mg | 30 | 2.46 | 1.32 | 30 | 2.04 | 1.25 | | 0.42 | [-0.23; 1.07] | 11.9% |
| Random effects model | | | | | | | | | -0.74 | [-1.08; -0.40] | 100.0% |

Heterogeneity: $I^2 = 99\%$, $\tau^2 = 0.1414$, $p < 0.01$

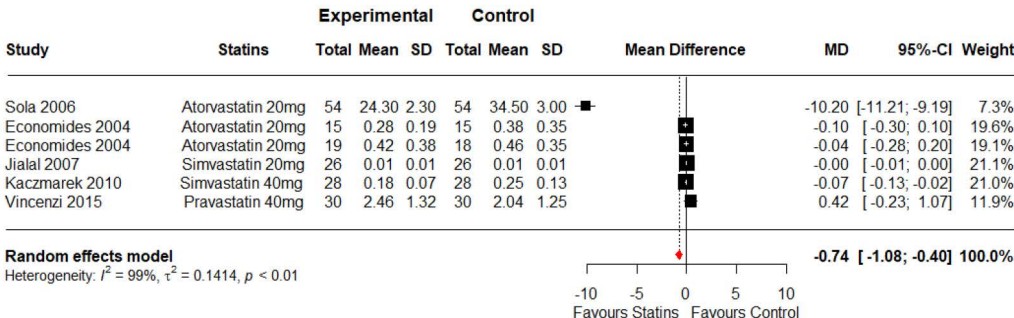

-10 -5 0 5 10
Favours Statins  Favours Control

## C. CRP

|  |  | Experimental | | | Control | | | Mean Difference | MD | 95%-CI | Weight |
|---|---|---|---|---|---|---|---|---|---|---|---|
| Study | Statins | Total | Mean | SD | Total | Mean | SD | | | | |
| Liu 2009 | Atorvastatin 10mg | 33 | 5.24 | 1.79 | 32 | 6.62 | 2.58 | | -1.38 | [-2.46; -0.30] | 9.7% |
| Tan 2002 | Atorvastatin 10mg | 39 | 1.62 | 1.97 | 41 | 1.74 | 1.53 | | -0.12 | [-0.90; 0.66] | 11.1% |
| Vernaglione 2004 | Atorvastatin 10mg | 16 | 7.25 | 3.25 | 17 | 8.50 | 3.50 | | -1.25 | [-3.55; 1.05] | 4.9% |
| Puurunen 2013 | Atorvastatin 20mg | 19 | 2.80 | 6.00 | 19 | 1.80 | 2.10 | | 1.00 | [-1.86; 3.86] | 3.7% |
| Kitas 2019 | Atorvastatin 40mg | 1504 | 3.20 | 3.81 | 1498 | 4.19 | 4.46 | | -0.99 | [-1.29; -0.69] | 12.9% |
| Suleiman 2012 | Atorvastatin 80mg | 62 | 2.20 | 2.50 | 63 | 4.90 | 19.34 | | -2.70 | [-7.52; 2.12] | 1.6% |
| Kaczmarek 2010 | Simvastatin 40mg | 28 | 3.62 | 4.47 | 28 | 3.52 | 5.03 | | 0.10 | [-2.39; 2.59] | 4.4% |
| Heljic 2009 | Simvastatin 40mg | 45 | 4.65 | 1.71 | 50 | 5.08 | 2.04 | | -0.43 | [-1.18; 0.32] | 11.2% |
| de Lemos 2004 | Simvastatin 40/80mg | 2265 | 2.10 | 2.07 | 2232 | 2.63 | 2.52 | | -0.53 | [-0.66; -0.40] | 13.2% |
| Zanetti 2020 | Rosuvastatin 10mg | 21 | 15.60 | 33.00 | 21 | 31.10 | 15.00 | | -15.50 | [-31.00; 0.00] | 0.2% |
| Broch 2014 | Rosuvastatin 10mg | 36 | 2.67 | 3.00 | 35 | 4.83 | 6.85 | | -2.16 | [-4.63; 0.31] | 4.5% |
| Peters 2010 | Rosuvastatin 40mg | 702 | 1.70 | 2.50 | 282 | 3.40 | 10.70 | | -1.70 | [-2.96; -0.44] | 8.8% |
| Ichihara 2002 | Fluvastatin 20mg | 12 | 2.30 | 1.70 | 10 | 10.10 | 3.50 | | -7.80 | [-10.17; -5.43] | 4.7% |
| Ichihara 2002 | Fluvastatin 20mg | 12 | 2.60 | 1.60 | 10 | 9.70 | 2.60 | | -7.10 | [-8.95; -5.25] | 6.3% |
| Vincenzi 2015 | Pravastatin 40mg | 30 | 2.71 | 5.51 | 30 | 4.03 | 7.96 | | -1.32 | [-4.78; 2.14] | 2.7% |
| Random effects model | | | | | | | | | -1.58 | [-2.23; -0.94] | 100.0% |

Heterogeneity: $I^2 = 87\%$, $\tau^2 = 0.8106$, $p < 0.01$

-30 -20 -10 0 10 20 30
Favours Statins  Favours Control

**Fig 2. Meta-analysis showing the overall effect of different statin types versus placebo for (A) IL-6, (B) TNF-α, and (C) CRP.** SD = standard deviation, MD = mean difference, CI = confidence interval, CRP = C-reactive protein.

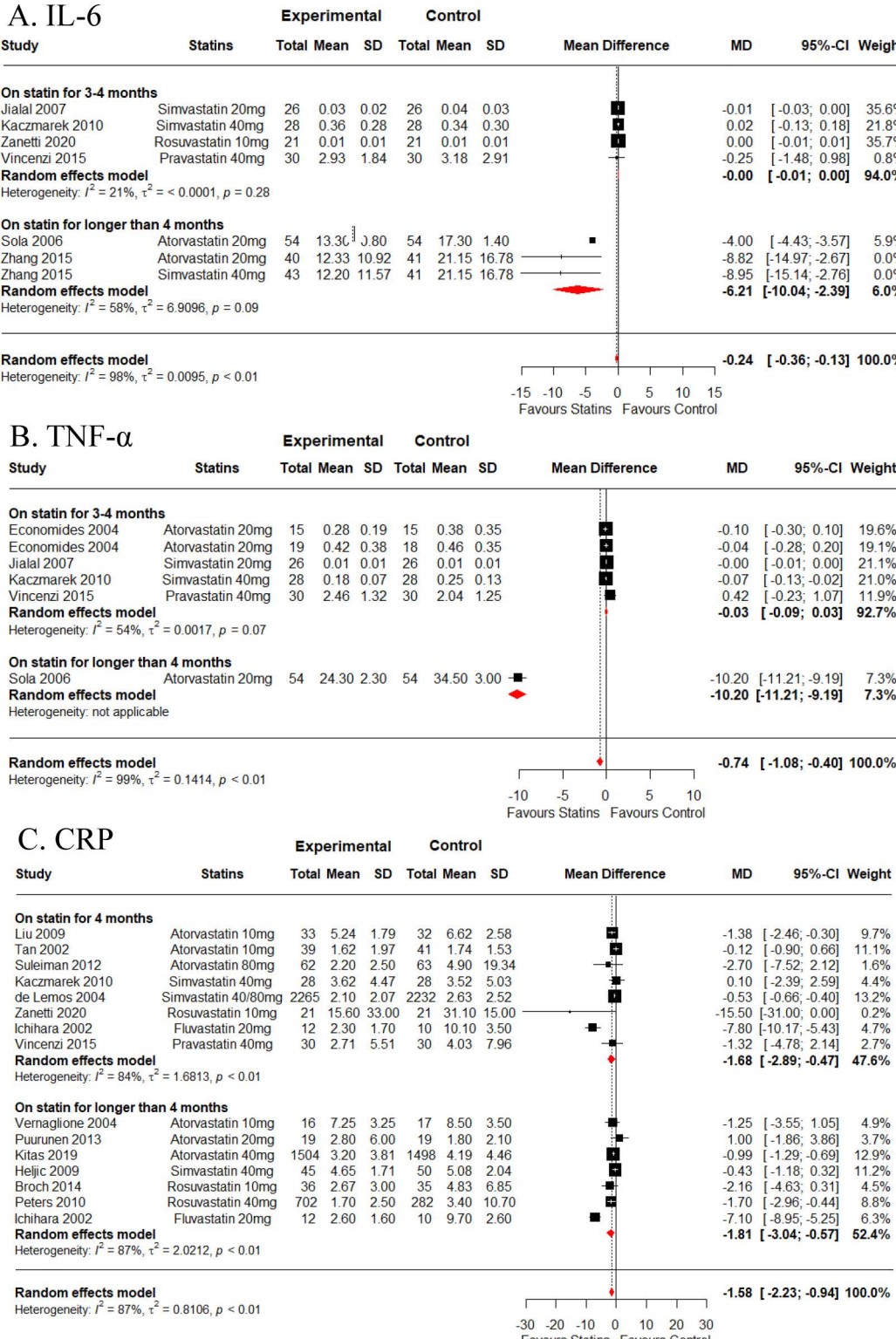

**Fig 3. Stratified meta-analysis for treatment duration effects of statins versus placebo subdivided into 3-4 months and longer than 4 months of treatment for (A) IL-6, (B) TNF-α, and (C) CRP.** SD = standard deviation, MD = mean difference, CI = confidence interval, CRP = C-reactive protein.

**Table 2. Analysis of randomized controlled trials stratified based on treatment duration.**

| Groups | Outcomes | Criteria | Studies (n) | Intervention (n) | Control (n) | Mean Differences (95 CI) | I² (95 CI) | P-value | P-Egger test | Certainty of the Evidence (GRADE) |
|---|---|---|---|---|---|---|---|---|---|---|
| Treatment duration | >4 months | | | | | | | | | |
| | IL-6 | Overall | 3 | 137 | 136 | -6.21 [-10.04; -2.39] | 58.1 [0.0; 88.1] | 0.092 | 0.006 | ⊕⊕⊖⊖ Low b,d |
| | | Atorvastatin | 2 | 94 | 95 | -5.39 [-9.68; -1.11] | 57.4 [0.0; 89.8] | 0.125 | NC | ⊕⊕⊖⊖ Low b,d |
| | | Simvastatin | 1 | 43 | 41 | -8.95 [-15.14; -2.76] | NC | NC | NC | |
| | TNF-α | Overall | 1 | 54 | 54 | -10.20 [-11.20; -9.19] | NC | NC | NC | |
| | | Atorvastatin | 1 | 54 | 54 | -10.20 [-11.20; -9.19] | NC | NC | NC | |
| | CRP | Overall | 7 | 2334 | 1911 | -1.81 [-3.04; -0.57] | 87.4 [76.4; 93.3] | <0.001 | 0.416 | ⊕⊕⊕⊖ Moderate a |
| | | Atorvastatin | 3 | 1539 | 1534 | -0.97 [-1.27; -0.68] | 0.0 [0.0; 89.6] | 0.387 | 0.596 | ⊕⊕⊖⊖ Low a,b |
| | | Simvastatin | 1 | 45 | 50 | -0.43 [-1.18; 0.32] | NC | NC | NC | |
| | | Rosuvastatin | 2 | 738 | 317 | -1.80 [-2.92; -0.67] | 0.0 [0.0; 0.0] | 0.745 | NC | ⊕⊕⊖⊖ Low a,b |
| | | Fluvastatin | 1 | 12 | 10 | -7.10 [-8.95; -5.25] | NC | NC | NC | |
| | hs-CRP | Overall | 6 | 9159 | 9157 | -1.58 [-1.95; -1.21] | 38.5 [0.0; 75.5] | 0.149 | 0.210 | ⊕⊕⊕⊖ Moderate b |
| | | Atorvastatin | 3 | 110 | 111 | -1.85 [-2.35; -1.36] | 0.0 [0.0; 89.6] | 0.936 | 0.508 | ⊕⊕⊖⊖ Low b,d |
| | | Simvastatin | 2 | 148 | 145 | -1.11 [-3.78; 1.56] | 53.6 [0.0; 88.5] | 0.142 | NC | ⊕⊖⊖⊖ Very low a,b,d |
| | | Rosuvastatin | 1 | 8901 | 8901 | -1.30 [-1.38; -1.22] | NC | NC | NC | |
| Treatment duration | 3-4 months | | | | | | | | | |
| | IL-6 | Overall | 4 | 105 | 105 | -0.00 [-0.01; 0.01] | 20.9 [0.0; 87.9] | 0.284 | 0.652 | ⊕⊖⊖⊖ Very low a,b,c |
| | | Simvastatin | 2 | 54 | 54 | -0.01 [-0.03; -0.00] | 0.0 [0.0; 0.0] | 0.618 | | ⊕⊖⊖⊖ Very low a,b,d |
| | | Pravastatin | 1 | 30 | 30 | -0.25 [-1.48; 0.98] | NC | NC | NC | |
| | | Rosuvastatin | 1 | 21 | 21 | -0.00 [-0.00; 0.00] | NC | NC | NC | |
| | TNF-α | Overall | 5 | 118 | 117 | -0.03 [-0.09; 0.03] | 53.7 [0.0; 83.0] | 0.071 | 0.516 | ⊕⊕⊖⊖ Low a,b |
| | | Atorvastatin | 2 | 34 | 33 | -0.07 [-0.23; 0.08] | 0.0 [0.0; 0.0] | 0.706 | NC | ⊕⊖⊖⊖ Very low a,b,d |
| | | Simvastatin | 2 | 54 | 54 | -0.03 [-0.10; 0.04] | 83.4 [31.1; 96.0] | 0.014 | NC | ⊕⊖⊖⊖ Very low a,b,d |
| | | Pravastatin | 1 | 30 | 30 | 0.42 [-0.23; 1.07] | NC | NC | NC | |
| | CRP | Overall | 8 | 2490 | 2457 | -1.68 [-2.89; -0.47] | 84.2 [70.4; 91.5] | <0.001 | 0.168 | ⊕⊕⊖⊖ Low a,b |

*(Continued)*

| Groups | Outcomes | Criteria | Studies (n) | Intervention (n) | Control (n) | Mean Differences (95 CI) | I² (95 CI) | P-value | P-Egger test | Certainty of the Evidence (GRADE) |
|---|---|---|---|---|---|---|---|---|---|---|
| | | Atorvastatin | 3 | 134 | 136 | -0.79 [-1.92; 0.34] | 52.3 [0.0; 86.3] | 0.122 | 0.544 | ⊕⊖⊖⊖ Very low [a,b,c] |
| | | Simvastatin | 2 | 2293 | 2260 | -0.53 [-0.66; -0.39] | 0.0 [0.0-0.0] | 0.620 | NC | ⊕⊖⊖⊖ Very low[a,b,d] |
| | | Pravastatin | 1 | 30 | 30 | -1.32 [-4.78; 2.14] | NC | NC | NC | |
| | | Rosuvastatin | 1 | 21 | 21 | -15.50 [-31.00; 0.00] | NC | NC | NC | |
| | | Fluvastatin | 1 | 12 | 10 | -7.80 [-10.17; -5.43] | NC | NC | NC | |
| | hs-CRP | Overall | 10 | 273 | 278 | -0.37 [-0.72; -0.01] | 94.7 [92.9; 96.4] | <0.001 | 0.102 | ⊕⊕⊖⊖ Low [a,b] |
| | | Atorvastatin | 8 | 225 | 231 | -0.32 [-0.76; 0.12] | 94.5 [91.4; 96.6] | <0.001 | 0.204 | ⊕⊕⊖⊖ Low [a,b] |
| | | Simvastatin | 2 | 48 | 47 | -0.65 [-0.98; -0.33] | 8.9 [NC; NC] | 0.294 | NC | ⊕⊖⊖⊖ Very low [a,b,c] |

n = Total numbers; NC = Not computable; IL-6 = Interleukin 6; TNF-α = Tumor necrosis factor alpha; CRP = C-reactive protein; hs-CRP = High-sensitivity C-reactive protein.

GRADE Working Group grades of evidence:

- High certainty: we are very confident that the true effect lies close to the effect estimate.

- Moderate certainty: we are moderately confident in the effect estimate: the true effect is likely to be close to the estimate of the effect, but there is a possibility that it is substantially different.

- Low certainty: our confidence in the effect estimate is limited: the true effect may be substantially different from the estimate of the effect.

- Very low certainty: we have very little confidence in the effect estimate: the true effect is likely to be substantially different from the estimate of effect.

a: serious inconsistency (high heterogeneity, wide confidence intervals); b: serious imprecision (optimal information size (OIS) not met, small number of studies); c: serious indirectness (studied populations differed with the populations recommended); d: serious risk of bias (ROB) (if high ROB presented in half or less than half of included studies).

14240 participants were on statins treatment arm and 13786 were on placebo control. Pooled data estimates showed that statins significantly reduced LDL-C (MD = -0.98 mmol/L [95% CI, -1.17 to -0.78], I² = 98.1%, p < 0.001) as depicted in Fig 4A and S6 Table. Collective analysis showed that rosuvastatin emerged as the most potent LDL-C reducer compared to other statins (MD = -1.48 mmol/L [95% CI, -1.86 to -1.10], I² = 97.9%, p < 0.001), shortly followed by atorvastatin (MD = -1.10 mmol/L [95% CI, -1.31 to -0.88], I² = 94.1%, p < 0.001), and simvastatin (MD = -0.78 mmol/L [95% CI, -1.16 to -0.39], I² = 98.8%, p < 0.001), as displayed in S6 Table. Notably, extended treatment duration (> 4 months) with rosuvastatin showcased the most potent capacity to decrease LDL-C concentration, as shown in Fig 4B and S7 Table. Regarding the effects of statins on triglyceride (TG), twenty-seven studies were included with a total of 26823 participants (13428 on statins and 13395 on placebo) (S6 Table). In this meta-analysis, pooled data estimates showed that statins significantly reduce TG levels (MD = -0.20 mmol/L [95% CI, -0.29 to -0.11], I² = 91.5%, p < 0.001). Pooled analysis from twenty-seven studies with a total population of 9133 participants (4583 on statins and 4550 on placebo) revealed that statins significantly decreased total cholesterol (TC) (MD = -1.01 mmol/L [95% CI, -1.24 to -0.78], I² = 97.2%, p < 0.001) with atorvastatin (MD = -1.24 mmol/L [95% CI, -1.55 to -0.93], I² = 95.7%, p < 0.001) showing the most potency in reducing TC, followed by rosuvastatin (MD = -1.04 mmol/L [95% CI, -1.60 to -0.49], I² = 61.9%, p = 0.105) and simvastatin (MD = -0.88 mmol/L [95% CI, -1.29 to -0.46], I² = 97%, p < 0.001) as outlined in S6 Table. All NRS showed a reduction in LDL-C levels, as summarized in S2 Table.

## A. LDL-C

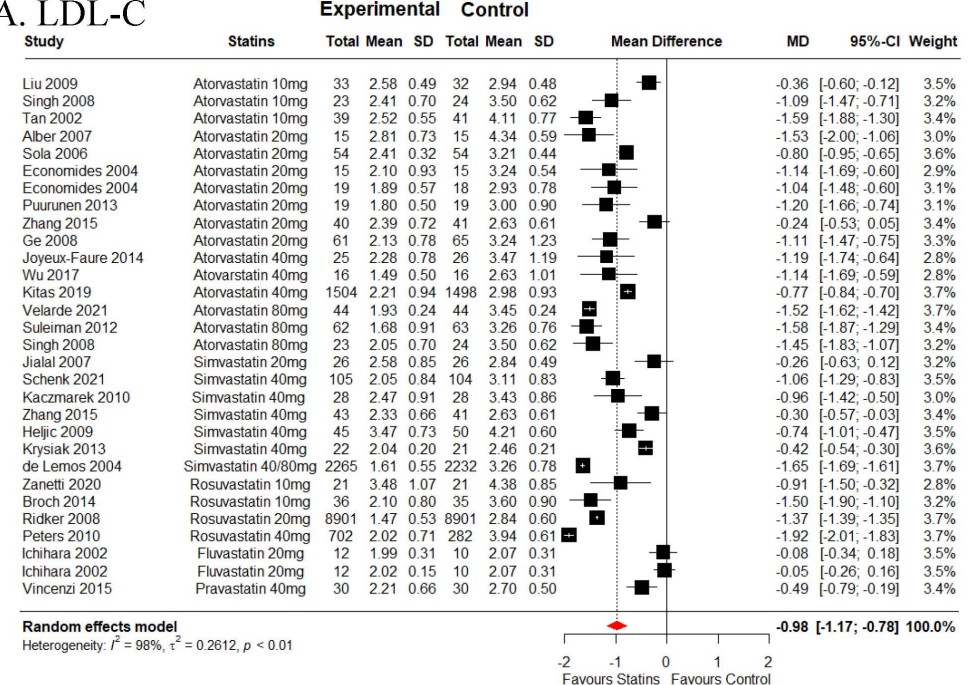

## B. LDL-C

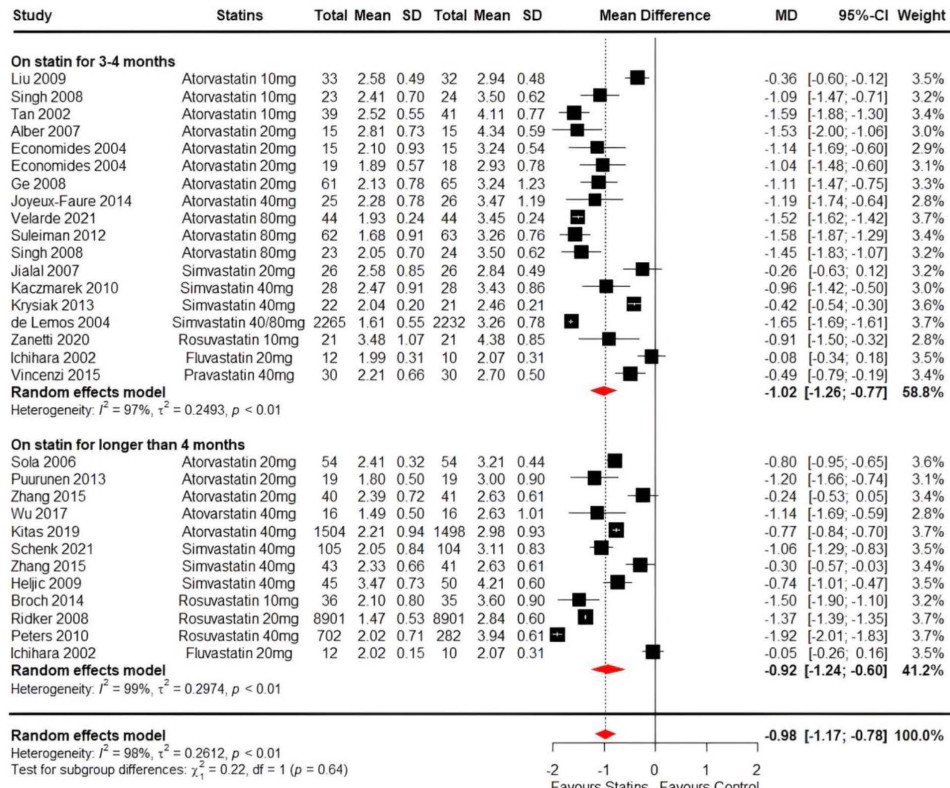

**Fig 4. LDL-C meta-analysis showing in (A) the overall effects of different statin types versus placebo and (B) stratified by statin treatment duration for 3-4 months and longer than 4 months.** SD = standard deviation, MD = mean difference, CI = confidence interval, LDL-C: low-density lipoprotein cholesterol.

## Publication bias

The result of Egger's test (Table 1) and funnel plots did not show any evidence of publication bias in the included studies for IL-6 and TNF-α (Fig 5A-5B). However, some publication bias was observed in studies evaluating CRP (p for Egger's test = 0.048) and LDL-C levels (p for Egge's test = 0.022) (Fig 5C-5D) as summarized in S11 Table.

## Discussion

This systematic review and meta-analysis provide a comprehensive assessment of the effectiveness of statins in mitigating systemic inflammation and altering lipid profiles during chronic illnesses, with statin administration lasting 3 months or longer. This study aimed to shed light on the potential anti-inflammatory benefits of statin administration by encompassing a wide array of chronic diseases such as CVD, COPD, schizophrenia, and chronic renal failure. Through meticulous

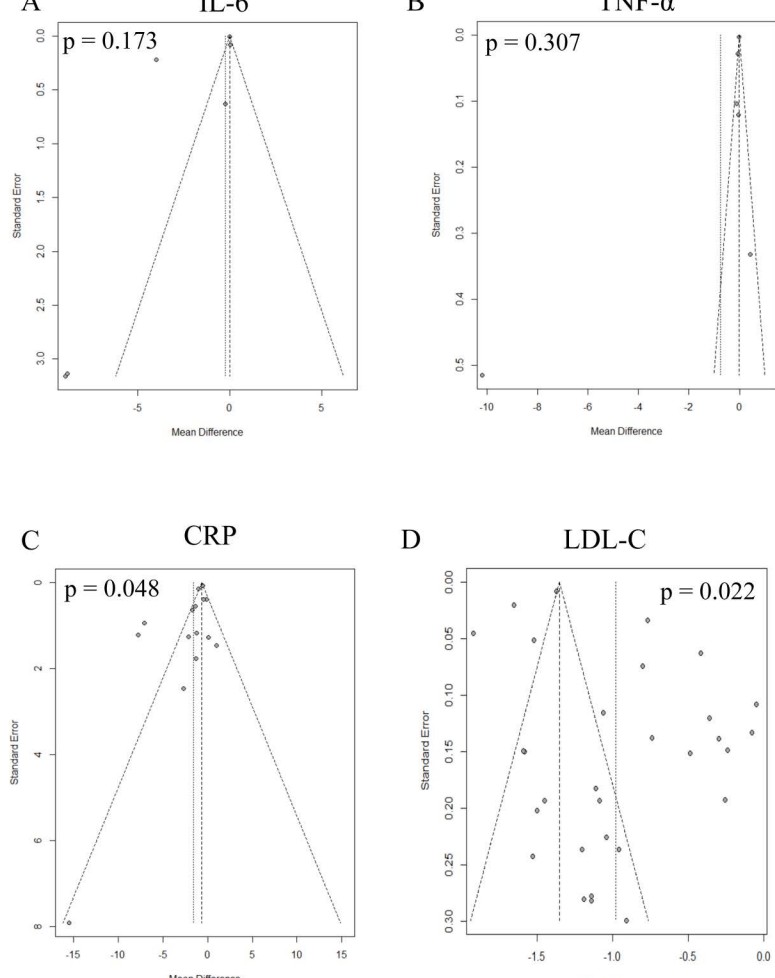

**Fig 5. Funnel plots of studies included in the meta-analysis assessing (A) IL-6, (B) TNF-α and (C) CRP inflammatory biomarkers, and (D) LDL-C lipid profile.** For each outcome, mean difference which was calculated for each study using random effect models with inverse variance method (horizontal axis) is plotted against its standard error (vertical axis) and represented by a dot. When the dots distribute symmetrically in a funnel shape, this implies an absence of bias. A p-value <0.05 (Egger's test) indicates significant publication bias. CRP = C-reactive protein, LDL-C: low-density lipoprotein cholesterol.

screening of literature across four distinct databases - PubMed, Web of Science, Scopus, and Cochrane - 27 RCT articles met the initial eligibility criteria. The findings from this meta-analysis highlight the significant impact of statin therapy in reducing systemic inflammatory markers, including IL-6, TNF-α, and CRP, across various chronic diseases (low to moderate-certainty of evidence). Analyzing data from multiple randomized controlled trials (RCTs), the study observed a statistically significant reduction in IL-6 (MD = -0.24 ng/dL, low-certainty evidence), TNF-α (MD = -0.74 ng/dL, low-certainty evidence), and CRP (MD = -1.58 mg/L, moderate-certainty evidence) among participants receiving statins compared to those on placebo. Atorvastatin (20 mg) emerged as particularly effective in reducing IL-6 (MD = -5.39 ng/dL, low-certainty evidence) and TNF-α (MD = -3.32 ng/dL, low-certainty evidence), highlighting its potency in modulating these inflammatory markers (Table 1). Additionally, extended statin administration beyond four months demonstrated heightened efficacy in lowering IL-6 and TNF-α levels (Fig 3A-3B), suggesting a potential temporal relationship between treatment duration and anti-inflammatory impact. However, data supporting the heightened efficacy for lowering TNF-α levels primarily originates from a single study (Fig 3B). IL-6, evaluated across seven RCTs involving 483 participants, and TNF-α, assessed in six RCTs involving 343 participants, both demonstrated substantial heterogeneity (IL-6: $I^2 = 98.3\%$; TNF-α: $I^2 = 98.8\%$), indicating variability in study outcomes despite overall significant reductions. Fifteen studies were included in CRP analysis with overall heterogeneity of $I^2 = 86.5\%$ suggestive of high variety in included studies (Fig 2C and Table 1). The observed heterogeneity suggests that factors such as participant demographics, differences in study populations, baseline disease severity, treatment durations, and statin dosage might influence treatment outcomes. Notably, CRP levels were also reduced by statin treatment, especially by fluvastatin followed by rosuvastatin (Table 1). It's noteworthy that CRP, a non-specific marker of inflammation, is associated with CVD pathogenesis [46]. Data generated in this meta-analysis contrasts with those of the study conducted by Balk *et al.*, 2003, where fluvastatin was observed to have the least impact on CRP reduction compared to other statins such as atorvastatin, simvastatin, and pravastatin [47]. However, it's important to acknowledge that this discrepancy in results could be attributed to the limited number of available studies that evaluated the efficacy of fluvastatin specifically in the context of chronic illness. While the existing evidence supports that statin treatment lowers systemic inflammatory markers levels, additional RCTs are still needed to delve deeper into the effects of statins on various inflammatory markers in chronic illnesses. The ultimate objective of these trials would be to establish a comprehensive understanding of how statin treatment influences a range of inflammatory markers and whether these effects translate into tangible improvements in disease-specific clinical outcomes.

Thirty studies encompassing over 28,000 participants demonstrated that statin therapy significantly reduced LDL-C levels (MD = -0.98 mmol/L), with substantial heterogeneity ($I^2 = 98.1\%$). Rosuvastatin emerged as the most effective LDL-C reducer (MD = -1.48 mmol/L), followed by atorvastatin (MD = -1.10 mmol/L) and simvastatin (MD = -0.78 mmol/L). This efficacy trend has been previously reported by Schachter M. *et al.*, 2005, which showed that rosuvastatin led to a 63% reduction in LDL-C, followed by atorvastatin with 50% and simvastatin with 41% [48]. Extended treatment duration (> 4 months) with rosuvastatin showed the greatest potency in lowering LDL-C concentrations (Fig 4B, S7 Table). These findings underscore the differential efficacy of statins in reducing LDL-C, reflecting their varying pharmacokinetic profiles and potency in inhibiting cholesterol synthesis via the mevalonate pathway. Twenty-seven studies involving over 26,800 participants demonstrated a significant reduction in TG levels with statin therapy (MD = -0.20 mmol/L, $I^2 = 91.5\%$). Similarly, statins significantly decreased TC levels across twenty-seven studies (MD = -1.01 mmol/L, $I^2 = 97.2\%$). Atorvastatin exhibited the most potent effect on TC reduction (MD = -1.24 mmol/L), followed by rosuvastatin (MD = -1.04 mmol/L) and simvastatin (MD = -0.88 mmol/L). The variability in efficacy among statins for TG and TC reductions may stem from differences in lipid metabolism pathways targeted by each statin. The high heterogeneity observed across lipid outcomes underscores the importance of considering factors such as baseline lipid levels, statin dosage, and study duration when interpreting results. Heterogeneity may also reflect variations in participant characteristics and study methodologies. The stratification by treatment duration highlights the potential for longer-term statin therapy, particularly with rosuvastatin, to achieve more pronounced lipid-lowering effects. These findings have significant implications for clinical practice, suggesting that statins,

particularly rosuvastatin and atorvastatin, are effective in managing dyslipidemia across diverse patient populations. Tailoring statin therapy based on individual lipid profiles and treatment duration may optimize lipid control and reduce cardiovascular risk. Clinicians should consider the differential effects of statins on LDL-C, TG, and TC when selecting treatment regimens for patients with specific lipid management goals.

An important limitation of this meta-analysis is that the lipid-lowering effect sizes of the various statins are derived from highly heterogeneous studies with variable baseline lipids and data on LDL-C and TC changes were not available for all statins. These results should thus not be interpreted as direct comparisons of statin potency. In this study, we used random-effects models to account for variability among studies and provide more conservative estimates of the overall effect. Also, sensitivity analyses were performed to assess the robustness of the findings by excluding studies with skewed evidence, as shown in S11 Table. Despite these efforts, high heterogeneity remains a limitation that impacts the generalizability of the meta-analytic conclusion. To address publication bias, we employed comprehensive literature search strategies, including backward citation searching and trial registries. Additionally, we used statistical methods such as funnel plots and Egger's test to detect and adjust for potential bias. However, these methods have limitations, and the possibility of residual bias cannot be entirely ruled out. Notably, the low to moderate-certainty of evidence, as assessed using GRADE, being attributable to imprecision, inconsistency or high risk of bias of some trials suggests that further research may change these results.

In conclusion, this meta-analysis provides robust evidence supporting the anti-inflammatory effects of statins across various chronic diseases. Atorvastatin, fluvastatin, and rosuvastatin emerged as particularly effective in reducing IL-6, TNF-α, and CRP levels, respectively. Based on several assumptions, the effect size of statins on inflammatory markers in this meta-analysis was small to moderate and of limited clinical significance for statins overall. However, atorvastatin showed a likely large and clinically relevant impact on IL-6 and TNF-a, while fluvastatin had a very large effect on CRP, and rosuvastatin exhibited a moderate-to-large effect on CRP. The greater anti-inflammatory efficacy of rosuvastatin and atorvastatin is primarily attributed to their minimal first-pass effect and relatively slow hepatic elimination. In contrast, other statins such as pravastatin undergo extensive first-pass metabolism and are rapidly inactivated by the liver before reaching systemic circulation. As a result, atorvastatin and rosuvastatin have longer elimination half-lives of 14 and 19 hours, respectively, while pravastatin's elimination half-life is only 2 hours. These findings have significant clinical implications for the management of chronic diseases characterized by inflammation, such as cardiovascular disease and chronic renal failure. Incorporating statin therapy into treatment protocols not only addresses lipid abnormalities but also potentially mitigates inflammatory processes implicated in disease progression. Clinicians may consider the differential efficacy of statins in inflammation modulation when tailoring treatment strategies for individual patients.

## Conclusion

This study reaffirms the existing body of literature that underscores the superior efficacy of rosuvastatin in improving lipid profiles. Additionally, statins proved to exert pleiotropic properties in various disease settings through their ability to modulate secreted inflammatory mediators like CRP, IL-6, and TNF-α. This meta-analysis provides valuable insights into the potential utility of statins as either primary interventions or adjunctive host-directed therapies, offering the promise of reducing inflammation and ultimately enhancing disease outcomes in the context of chronic infections. This aligns with observations from the Pravastatin or Atorvastatin Evaluation and Infection Therapy-Thrombolysis in Myocardial Infarction 22 (PROVE IT-TIMI 22) study, where improved clinical outcomes in patients were associated with reduced CRP levels independent of lipid profiles [49]. However, it is important to emphasize the need for additional rigorous randomized controlled trials (RCTs) to comprehensively explore and characterize the effects of statins on a wider spectrum of inflammatory biomarkers and clinical outcomes, thereby providing a more holistic understanding of their potential to influence disease progression and pathogenesis.

## Supporting information

**S1 Table. Reason for exclusion of all 6514 screened articles.** RCT = Randomized Controlled Trial.
(XLSX)

**S2 Table. Summary of non-randomized studies.** n = participant number, LDL-C = Low-Density Lipoprotein-Cholesterol, HDL = High-Density Lipoprotein, TG = Triglycerides, TC = Total Cholesterol, CRP = C-Reactive Protein, IL = Interleukin, sCD = soluble Cluster of Differentiation, TNF-α = Tumour Necrosis Factor-alpha, NR = Not Reported.
(XLSX)

**S3 Table. Summary of randomized controlled trials.** Keys: RCT = Randomized controlled trials; n = total number; yrs = years; NR = not reported; mmol = millimoles per litre; mg/L = milligrams per liter; ng/dL = nanograms per deciliter; ∗ = Studies with subgroup analysis.
(XLSX)

**S4 Table. Jadad Score of randomized controlled trials.** * = Excluded studies (outcome not compatible with meta-analysis) (n = 5), Scoring: 1 = Reported, 0 = not reported, Quality scoring: 0 ≥ 2 low, 3 ≥ 5 high.
(XLSX)

**S5 Table. Version 2 of the Cochrane risk-of-bias tool for randomized controlled trials.** * = Excluded studies (outcome not compatible with meta-analysis). (n = 5).
(XLSX)

**S6 Table. Overall analysis of randomized controlled trials.**
(DOCX)

**S7 Table. Analysis of randomized controlled trials stratified based on treatment duration.**
(DOCX)

**S8 Table. Methodological items for non-randomized studies (MINORS).** Scoring: 2 = Reported and Adequate, 0 = Not Reported.
(XLSX)

**S9 Table. Newcastle-Ottawa scale.** Scoring: * = Reported.
(XLSX)

**S10 Table. Summary table showing analysis of randomized controlled trials stratified based on age groups.** NC = Not Computable; CI = Confidence Interval; TC = Total Cholesterol; LDL-C = Low-Density Lipoprotein-Cholesterol; HDL = High-Density Lipoprotein; CRP = C-reactive protein; hs-CRP = high-sensitive C-reactive protein; IL-6 = Interleukin-6; TNF-α = Tumor Necrosis Factor-alpha.
(XLSX)

**S11 Table. Excluded skewed studies.** NC = Not Computable; CI = Confidence Interval; TC = Total Cholesterol; LDL-C = Low-Density Lipoprotein-Cholesterol; HDL = High-Density Lipoprotein; CRP = C-reactive protein; hs-CRP = high-sensitive C-reactive protein; IL-6 = Interleukin-6; TNF-α = Tumor Necrosis Factor-alpha.
(XLSX)

**S12 Table. PRISMA 2020 checklist.**
(PDF)

## Author contributions

**Conceptualization:** Solima Sabeel, Bongani Motaung, Mumin Ozturk, Friedrich Thienemann, Reto Guler.

**Data curation:** Solima Sabeel, Bongani Motaung, Kim A. Nguyen.

**Formal analysis:** Kim A. Nguyen, Andre Pascal Kengne.

**Funding acquisition:** Friedrich Thienemann, Reto Guler.

**Supervision:** Friedrich Thienemann, Reto Guler.

**Writing – original draft:** Solima Sabeel, Bongani Motaung.

**Writing – review & editing:** Mumin Ozturk, Sandra L. Mukasa, Karen Wolmarans, Dirk J. Blom, Karen Sliwa, Emmanuel Nepolo, Gunar Gunther, Robert J. Wilkinson, Claudia Schacht, Friedrich Thienemann, Reto Guler.

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
