## [Decision Letter · Decision Letter 0]

23 May 2024

PONE-D-24-09276Impact of statins as immune-modulatory agents on inflammatory markers in adults with chronic diseases: A systematic review and meta-analysisPLOS ONE

Dear Dr. Motaung,

Thank you for submitting your manuscript to PLOS ONE. After careful consideration, we feel that it has merit but does not fully meet PLOS ONE’s publication criteria as it currently stands. Therefore, we invite you to submit a revised version of the manuscript that addresses the points raised during the review process.

We look forward to receiving your revised manuscript.

Kind regards,

Timotius Ivan Hariyanto, M.D.

Academic Editor

PLOS ONE

Journal Requirements:

"This publication was produced by StatinTB which is part of the EDCTP2 programme supported by the European Union (grant number RIA2017T- 2004-StatinTB). This research was funded in whole, or in part, by the Wellcome Trust CIDRI-Africa [203135Z/16/Z]."

"This publication was produced by StatinTB which is part of the EDCTP2 programme supported by the European Union (grant number RIA2017T- 2004-StatinTB). This research was funded in whole, or in part, by the Wellcome Trust CIDRI-Africa [203135Z/16/Z]."

4. Please expand the acronym “EDCTP2” (as indicated in your financial disclosure) so that it states the name of your funders in full.

5. We note that this manuscript is a systematic review or meta-analysis; our author guidelines therefore require that you use PRISMA guidance to help improve reporting quality of this type of study. Please upload copies of the completed PRISMA checklist as Supporting Information with a file name “PRISMA checklist”.

**Additional Editor Comments:**

The reviewers have found merit in this article, however, they also found several major flaws within the manuscript that need further clarification and revision. Major points that need to be clarified:

1) Introduction: please make the introduction section not too long. Focus on the background to justify this study, literature gaps, and purpose of this study.

2) Methods: please explain why this manuscript has several differences with the protocol.

3) Discussion: please elaborate more the findings from this study and be more analytical to explain those results.

Please try to respond to the reviewers' comments one by one.

Reviewers' comments:

Reviewer's Responses to Questions

**Comments to the Author**

1. Is the manuscript technically sound, and do the data support the conclusions?

Reviewer #1: No

Reviewer #2: Yes

Reviewer #3: Yes

2. Has the statistical analysis been performed appropriately and rigorously? 

Reviewer #1: No

Reviewer #2: Yes

Reviewer #3: Yes

3. Have the authors made all data underlying the findings in their manuscript fully available?

Reviewer #1: Yes

Reviewer #2: Yes

Reviewer #3: Yes

4. Is the manuscript presented in an intelligible fashion and written in standard English?

Reviewer #1: Yes

Reviewer #2: Yes

Reviewer #3: Yes

5. Review Comments to the Author

Reviewer #1: Upon review, it is evident that the review protocol serves as a valuable document to demonstrate the authors' insight and plan for conducting the meta-analysis. However, the significant differences between the protocol and the present manuscript indicate a change in direction in the preparation of the meta-analysis, thereby increasing the risk of bias. Based on these observations, it is my professional opinion that the manuscript is not suitable for publication.

Reviewer #2: This is a well-written meta-analysis to assess the impact of statins as immune-modulatory agents on inflammatory markers. I have the following comments:

1. Please report if the high risk of bias in studies is an exclusion criterion in the methods or not.

2. Add a table to assess the quality of evidence of each outcome using the GRADE approach.

3. The results on lipid profile seems irrelevant to the aim of the study and these are already established, I would remove them.

4. Please define “chronic diseases”, also, a subgroup analysis according to the primary disease seems important in current study.

Reviewer #3: The manuscript presents a systematic review and meta-analysis on the anti-inflammatory effects of statins in chronic diseases, evaluating various inflammatory markers and lipid profiles. While the research is extensive and methodologically sound, there are several areas that need significant improvement to enhance the clarity, rigor, and impact of the findings.

Issues:

Introduction:

1. The introduction is lengthy and contains redundant information. It should be more concise, focusing on the rationale and objectives of the study.

2. The clinical importance of the anti-inflammatory effects of statins needs to be more explicitly stated.

Result:

The high heterogeneity (I2 > 75%) in many analyses raises concerns about the comparability of the included studies. This should be more thoroughly addressed in both the results and discussion sections.

Discussion:

1. The discussion is somewhat descriptive and needs to be more analytical. There should be a deeper exploration of why certain statins (e.g., atorvastatin) appear more effective than others.

2. The implications of the findings for clinical practice should be discussed in more detail. How might these results influence treatment guidelines?

3. The limitations section should be expanded. For example, the potential impact of publication bias and the high heterogeneity should be discussed more comprehensively.

6. PLOS authors have the option to publish the peer review history of their article (what does this mean? ). If published, this will include your full peer review and any attached files.

**Do you want your identity to be public for this peer review?** For information about this choice, including consent withdrawal, please see our Privacy Policy .

Reviewer #1: No

Reviewer #2: No

Reviewer #3: **Yes: ** Bayushi Eka Putra

---

## [Author Response · Author response to Decision Letter 1]

10 Jul 2024

We thank all the reviewers for the positive feedback that we received. We have addressed all the concerns that you raised during the review process.

---

## [Decision Letter · Decision Letter 1]

12 Aug 2024

PONE-D-24-09276R1Impact of statins as immune-modulatory agents on inflammatory markers in adults with chronic diseases: A systematic review and meta-analysisPLOS ONE

Dear Dr. Motaung,

Thank you for submitting your manuscript to PLOS ONE. After careful consideration, we feel that it has merit but does not fully meet PLOS ONE’s publication criteria as it currently stands. Therefore, we invite you to submit a revised version of the manuscript that addresses the points raised during the review process.

We look forward to receiving your revised manuscript.

Kind regards,

Timotius Ivan Hariyanto, M.D.

Academic Editor

PLOS ONE

Journal Requirements:

Reviewers' comments:

Reviewer's Responses to Questions

**Comments to the Author**

1. If the authors have adequately addressed your comments raised in a previous round of review and you feel that this manuscript is now acceptable for publication, you may indicate that here to bypass the “Comments to the Author” section, enter your conflict of interest statement in the “Confidential to Editor” section, and submit your "Accept" recommendation.

Reviewer #1: All comments have been addressed

Reviewer #2: All comments have been addressed

2. Is the manuscript technically sound, and do the data support the conclusions?

Reviewer #1: Yes

Reviewer #2: Yes

3. Has the statistical analysis been performed appropriately and rigorously? 

Reviewer #1: Yes

Reviewer #2: Yes

4. Have the authors made all data underlying the findings in their manuscript fully available?

Reviewer #1: Yes

Reviewer #2: Yes

5. Is the manuscript presented in an intelligible fashion and written in standard English?

Reviewer #1: Yes

Reviewer #2: Yes

6. Review Comments to the Author

Reviewer #1: This meta-analysis effectively evaluates the impact of statins as immune-modulatory agents on inflammatory markers. However, there are several areas that require attention before considering this manuscript for publication:

1. The protocol mentioned a search in the ScienceDirect database, but the final manuscript does not reflect this. It would be beneficial to provide a rationale for deviating from the protocol in this regard.

2. Reviewer 1 raised a question about the difference between the protocol and the manuscript, particularly regarding the inclusion of non-RCT studies. It is important to justify the inclusion of non-RCT studies and explain how they contributed to the findings.

3. I strongly recommend that the authors discuss the clinical application of statin use, including the magnitude of effect size (small, moderate, and large) and the minimum clinically important difference (MCID).

4. The protocol mentioned a subgroup analysis based on statin dosage, but this was not reflected in the forest plot and table. It is important to address why this subgroup analysis was not performed as planned.

Addressing these issues will significantly improve the manuscript and enhance its consideration for publication.

Reviewer #2: A meta-analysis to assess the impact of statins as immunomodulatory agents on inflammatory markers. The authors addressed all my comments and I have no further concerns, well done!

7. PLOS authors have the option to publish the peer review history of their article (what does this mean? ). If published, this will include your full peer review and any attached files.

**Do you want your identity to be public for this peer review?** For information about this choice, including consent withdrawal, please see our Privacy Policy .

Reviewer #1: No

Reviewer #2: No

---

## [Author Response · Author response to Decision Letter 2]

1 Nov 2024

All reviewer comments has been addressed in the rebuttal letter.

---

## [Decision Letter · Decision Letter 2]

26 Dec 2024

PONE-D-24-09276R2Impact of statins as immune-modulatory agents on inflammatory markers in adults with chronic diseases: A systematic review and meta-analysisPLOS ONE

Dear Dr. Motaung,

Thank you for submitting your manuscript to PLOS ONE. After careful consideration, we feel that it has merit but does not fully meet PLOS ONE’s publication criteria as it currently stands. Therefore, we invite you to submit a revised version of the manuscript that addresses the points raised during the review process.

We look forward to receiving your revised manuscript.

Kind regards,

Timotius Ivan Hariyanto, M.D.

Academic Editor

PLOS ONE

Journal Requirements:

Reviewers' comments:

Reviewer's Responses to Questions

**Comments to the Author**

1. If the authors have adequately addressed your comments raised in a previous round of review and you feel that this manuscript is now acceptable for publication, you may indicate that here to bypass the “Comments to the Author” section, enter your conflict of interest statement in the “Confidential to Editor” section, and submit your "Accept" recommendation.

Reviewer #1: All comments have been addressed

Reviewer #3: All comments have been addressed

2. Is the manuscript technically sound, and do the data support the conclusions?

Reviewer #1: Yes

Reviewer #3: Yes

3. Has the statistical analysis been performed appropriately and rigorously? 

Reviewer #1: No

Reviewer #3: Yes

4. Have the authors made all data underlying the findings in their manuscript fully available?

Reviewer #1: Yes

Reviewer #3: Yes

5. Is the manuscript presented in an intelligible fashion and written in standard English?

Reviewer #1: Yes

Reviewer #3: Yes

6. Review Comments to the Author

Reviewer #1: The manuscript presents an engaging and well-composed exploration of an intriguing topic. However, several areas warrant attention for improvement. Firstly, the exclusion of cancer and malignancy patients, including those with HIV or Arthritis, raises questions. It would be beneficial for the authors to consider including all patient groups and to define a category for health status participants in the subgroup analysis. This could provide valuable insights into the effect of statins on inflammation among these participants.

Secondly, I would appreciate clarification regarding the GRADE assessment method employed. Specifically, what approach was adopted to address heterogeneity? Was a null effect or small effect approach selected? This information appears to be missing.

Thirdly, the rationale behind utilizing the GRADE assessment should be elaborated upon. The authors should integrate the GRADE results into their discussion of findings to enhance the manuscript's depth.

Lastly, figures in the manuscript are of low quality and lack engagement. Improving these would significantly enhance the overall presentation of the work.

Reviewer #3: All questions have been addressed. No further confirmation needed. I recommend the paper to be accepted

7. PLOS authors have the option to publish the peer review history of their article (what does this mean? ). If published, this will include your full peer review and any attached files.

**Do you want your identity to be public for this peer review?** For information about this choice, including consent withdrawal, please see our Privacy Policy .

Reviewer #1: No

Reviewer #3: **Yes: ** Bayushi Eka Putra

---

## [Author Response · Author response to Decision Letter 3]

24 Feb 2025

We have included all 6514 articles that were screened for the meta-analysis and highlighted the reason for exclusion for each excluded article, now presented in our newly updated supplementary table 1 (S1_Table).

---

## [Decision Letter · Decision Letter 3]

15 Apr 2025

Impact of statins as immune-modulatory agents on inflammatory markers in adults with chronic diseases: A systematic review and meta-analysis

PONE-D-24-09276R3

Dear Dr. Motaung,

We’re pleased to inform you that your manuscript has been judged scientifically suitable for publication and will be formally accepted for publication once it meets all outstanding technical requirements.

Kind regards,

Timotius Ivan Hariyanto, M.D.

Academic Editor

PLOS ONE

Additional Editor Comments (optional):

Reviewers' comments:

Reviewer's Responses to Questions

**Comments to the Author**

1. If the authors have adequately addressed your comments raised in a previous round of review and you feel that this manuscript is now acceptable for publication, you may indicate that here to bypass the “Comments to the Author” section, enter your conflict of interest statement in the “Confidential to Editor” section, and submit your "Accept" recommendation.

Reviewer #1: All comments have been addressed

2. Is the manuscript technically sound, and do the data support the conclusions?

Reviewer #1: Yes

3. Has the statistical analysis been performed appropriately and rigorously? 

Reviewer #1: Yes

4. Have the authors made all data underlying the findings in their manuscript fully available?

Reviewer #1: Yes

5. Is the manuscript presented in an intelligible fashion and written in standard English?

Reviewer #1: Yes

6. Review Comments to the Author

Reviewer #1: Thank you for addressing my questions. I no longer have any further inquiries, and the manuscript is ready for acceptance.

7. PLOS authors have the option to publish the peer review history of their article (what does this mean? ). If published, this will include your full peer review and any attached files.

**Do you want your identity to be public for this peer review?** For information about this choice, including consent withdrawal, please see our Privacy Policy .

Reviewer #1: No

---

## [Editor Report · Acceptance letter]

PONE-D-24-09276R3

PLOS ONE

Dear Dr. Motaung,

I'm pleased to inform you that your manuscript has been deemed suitable for publication in PLOS ONE. Congratulations! Your manuscript is now being handed over to our production team.

Kind regards,

on behalf of

Dr. Timotius Ivan Hariyanto

Academic Editor

PLOS ONE